# Programmatic assessment of electronic Vaccine Intelligence Network (eVIN)

**Vandana Gurnani[1], Prem Singh[2], Pradeep Haldar[1], Mahesh Kumar Aggarwal[1], Kiran Agrahari[2], Satabdi Kashyap[2], Shreeparna Ghosh[2], Mrinal Kar Mohapatra[2], Ruma Bhargava[2], Partha Nandi[3], Pritu Dhalaria[2]***

**1** Immunization Division, Ministry of Health & Family Welfare, Government of India, New Delhi, India, **2** Immunization Technical Support Unit, Ministry of Health & Family Welfare, Government of India, New Delhi, India, **3** Mahatma Gandhi Medical College and Research Institute, Pondicherry, India

* pritu_dhalaria@in.jsi.com

**Data Availability Statement:** All data files are available from the DOI database (http://dx.doi.org/10.17632/ysnmgygmmn.1).

## Abstract

eVIN is a technology system that digitizes vaccine stocks through a smartphone application and builds the capacity of program managers and cold chain handlers to integrate technology in their regular work. To effectively manage the vaccine logistics, in 2015, this technology was rolled-out in 12 states of India. This study assessed the programmatic usefulness of eVIN implementation in the areas of vaccine utilization, vaccine stock and distribution management and documentation across selected cold chain points. A pre-post study design was used, where cold chain points (CCPs) were selected using two-stage sampling technique in eVIN states. Pre-post comparative analysis was carried out on the identified indicators using both primary and secondary data sources. The vaccine utilization data reflects that the utilization had reduced from 305.3 million doses in pre-eVIN period to 215.0 million doses in post-eVIN period across 12 eVIN states, resulting into savings of approximately 90 million doses of vaccines. Number of facilities having stock-out of any vaccine showed a significant reduction by 30.4% in post-eVIN period (p<0.001). There was a 4.0% drop in facilities reporting minimum stock of any vaccine after implementation of eVIN. Facilities with maximum stock of any vaccine had increased from 37.4% in pre-eVIN to 39.2% in post-eVIN. During the pre-eVIN period, only 38.6% facilities updated vaccine stock on a daily basis, while in post-eVIN period, 53.5% facilities updated vaccine stock on daily basis. The completeness of records in the vaccine stock registers, indent form and temperature logbook have been substantially improved in the post-eVIN period (p<0.001). eVIN had helped in streamlining the vaccine flow network and ensured equity through better vaccine management practices. It is a powerful contribution to strengthen the vaccine supply chain and management. Upscaling eVIN in the remaining states of India will be crucial in improving the efficacy of vaccines and cold chain management.

## Introduction

The Immunization Supply Chain and Logistics (ISCL) system, the backbone of any immunization program, plays a key role in improving the immunization coverage with quality and

**Funding:** The assessment was undertaken as a part of Gavi Targeted Country Assistance (TCA) support provided to Ministry of Health and Family Welfare, Government of India. These funds were utilized to collect the field level data. None of the authors received any monetary support for conducting this study from funders.The funders had no role in study design, data collection and analysis, decision to publish or preparation of the manuscript.

**Competing interests:** The authors have declared that no competing interests exist

equity through timely supply of safe and potent vaccines [1]. The Universal Immunization Programme (UIP) of India, one of the largest public health programmes in the world, caters to ~26 million new born and ~30 million pregnant women via 9 million sessions every year through 27,000 cold chain points [2]. The Immunization Supply Chain and Logistics (ISCL) system in the country has played a significant role in achieving the current coverage levels while dealing with the challenges of vaccine storage, distribution and management [3].

In India, several challenges related to the management of ISCL present themselves [4]. One among them is the maintance of records. Initially, for vaccine records and management, mainly four types of paper formats were used–comprehensive log book for every Cold Chain Equipment (CCE) to record temperature as well as details of repair and maintenance; vaccine stock register–issue and receipt; vaccine distribution register for immunization sessions, and vaccine and logistics indent form for recording temperature, maintenance and repair details of CCEs, for the vaccine stock, daily issue and logistics indent [5]. These formats require manual data entry, compilation and consolidation, which lead to a delay in real-time visibility of stock levels and temperatures. This leads to weak inventory and stock-flow record keeping practices and subsequently delays vaccine stock visibility [6]. Moreover, the temperature monitoring of CCE was largely dependent on the availability of a dedicated human resource at the cold chain point (CCP) [7]. This type of system maintenance poses serious challenges to the quality of recording, reporting and monitoring the temperature of CCEs.

In order to address the issues and strengthen the vaccine supply chain, in 2015, eVIN was introduced by the Ministry of Health and Family Welfare (MoHFW), Government of India to digitize the details of stock and storage temperature and enable real time visibility of vaccine inventories [8].

eVIN is a stock management technology that digitizes vaccine stocks through a smartphone application and builds the capacity of program managers and cold chain handlers to integrate the technology in their regular work. eVIN technology uses a smartphone, a web-based application, temperature loggers and cloud-based server to digitize vaccine stock inventory and storage temperature from every vaccine store and CCP located at peripheral government health facilities. It allows remote temperature monitoring for all CCE with automated alert mechanisms to alert designated staff in case if the temperature of equipment differs from the predefined range [9]. It's an integrated package of people, process and product [10]. For successful implementation of the eVIN system, the capacity development of cold chain handlers (people) is also ensured [11].

To evaluate the usefulness of eVIN implementation and to visualize its benefits and challenges, a programmatic assessment was undertaken. Although several immunization supply chain studies including the "National Effective Vaccine Management (EVM) Assessment" by the National Cold Chain Vaccine Management Resource Centre has been conducted on vaccine and cold chain management, no assessment has been conducted on the implementation of eVIN and its programmatic outcomes. This piece of work is part of a larger study on the 'Techno-economic Assessment of eVIN', conducted to disseminate learnings for a national scale up of the eVIN [12].

## Method

The assessment was conducted in 12 states (Assam, Chhattisgarh, Gujarat, Jharkhand, Manipur, Nagaland, Odisha, Bihar Himachal Pradesh, Madhya Pradesh, Rajasthan and Uttar Pradesh) where eVIN was launched initially. A pre-post comparison was used on key performance indicators for programmatic assessment. As obtaining data for one-year prior to implementation of eVIN was challenging, therefore, six months period was chosen as the

reference period for pre-eVIN phase. The pre-eVIN reference period was of six months for all CCPs, however it varied for different states and districts due to different timeframe of eVIN rollout. The specific duration of pre-eVIN phase are as mentioned: most of the CCPs in a district were covered either in between the period of April 2015 to September 2015 (13 districts), April 2016 to September 2016 (12 districts), October 2015-March 2016 (11 districts) and October 2011- March 2012 (1 district). The post eVIN reference period was from October 2017 to March 2018 for the assessment.

Minimum number of CCPs required for the study was calculated to be 502 considering 43% of PHCs reported instances of stock-out [13], 10% non-response rate and 1.2 design effect [14]. It was further increased to 617 CCPs in order to draw valid conclusions at the state levels.

Selection of CCPs in the eVIN states were done using two-stage sampling design. In the first stage, districts were selected followed by selection of CCPs in the second stage. In each eVIN state, number of sampled districts was decided based on Probability Proportion to Size of CCPs. In total 37 districts were selected using systematic random sampling technique after arranging the districts in ascending order based on the proportionate share of cold chain point in the total cold chain point in state. Further, CCPs were randomly selected in each of the selected district. A detailed methodology is available in the larger study document [12].

Quantitative data was obtained using structured questionnaire from CCPs pertaining to stock management, temperature monitoring, cold chain equipment, and documentation aspects of vaccine supply chain. The primary data for pre-eVIN phase was done from stock registers, vaccine distribution registers, temperature log books and other important registers. Completeness and accuracy were analyzed in the assessment. Completeness was seen of Indent form [15], vaccine stock register [16], and temperature log book. Accuracy was assessed through stock register and eVIN record, eVIN record and physical count. Computer Assisted Personal Interviewing (CAPI) technique was employed using tablets/mobiles for real-time data collection and data entry. Further, data quality assessment was done using survey CTO [17] and MS Excel. Data analysis was conducted in MS Excel and STATA 13 software.

A set of selected indicators fulfilling the objective are used in this paper for which pre-post data was collected to represent the programmatic assessment of eVIN (Table 1).

## Results

The study found that the utilization of vaccines reduced from 305.3 million doses in pre-eVIN period to 215.0 million doses in post-eVIN period, demonstrating 29.6% reduction in

**Table 1. List of programmatic assessment indicators, data sources and definition used.**

| Indicators | Data sources | Definition |
|---|---|---|
| Vaccine utilization | Secondary | Vaccine utilization focuses on the utilization of doses and pipeline stock savings of vaccine doses in pre- and post-period of eVIN implementation at the national level. Vaccine utilization in doses was computed as: *Vaccine Utilization in million doses = (Opening doses + Doses supplied from GMSD/ supplier)-Closing balance* |
| Stock Management | Primary & Secondary | This includes details on the events of stock-out, minimum and maximum stock or extent of excess stocks and missed opportunity [18] post eVIN. |
| Vaccine distribution | Primary | The effectiveness was assessed through lesser replenishment time, complete order fill rate [19], minimal expiry of vaccines at stores [20] and CCP to CCP sharing of vaccines. |
| Vaccine management practices | Primary | It included record keeping practices and vaccine stock updating duration |
| Documentation | Primary | Assessed through checking completeness and accuracy of records.<br>Completeness: Data completeness was categorized as a) more than 90%, b) between 80 to 90%, and c) below 80%. Above 90% completeness indicated less than or equal to 10 instances missed for critical indicators; 80 to 90% indicated less than or equal to 20 instances missed and below 80% indicated more than 20 instances missed.<br>Accuracy: It was done by tallying vaccine-wise stock kept in Ice Lined Refrigerator (ILR) and information recorded in stock register and eVIN record. Two different comparisons were carried out: One between stock register and eVIN record and another between eVIN record and physical count. |

utilization of doses. The facilities witnessing stock-out of any antigen had also reduced from 37.8% in pre-eVIN period to 26.3% in post-eVIN period. Number of facilities having stock-out of any vaccine showed a significant reduction of 30.4% in post-eVIN period (p<0.001). In the areas of minimum and maximum stock management situations, it was observed that the instances of minimum stock of any antigen was occurred at 49.8% facilities in the pre-eVIN period, compared with 47.8% facilities in the post-eVIN period. The maximum stock of any antigen had increased from 37.4% in pre-eVIN to 39.2% in post-eVIN period, indicated 4.8% increase in facilities witnessing events of maximum stock. The analysis of missed opportunities due to reduction in stock-outs revealed that after implementation of eVIN, fewer beneficiaries were getting omitted due to stock-out as compared in the pre-eVIN period. The highest reduction was observed in DPT (reduced by 70.0%) (p<0.001), and lowest in BCG (reduced by 6.4%) and almost no change in Hep-B (reduced by 0.2%) after eVIN implementation. Furthermore, mean number of days (of vaccine expiry) had reduced from 428 in pre-eVIN to 384 in the post-eVIN period at CCP level signifying that the 'First Expiry First Out' (FEFO) was being practiced. Overall improvement in utilizing the Government of India registers and reduction in usage of loose papers across facilities after the introduction of eVIN was statistically significant (p<0.001). For stock management, the updating of vaccine stock register daily had improved from 38.6% in pre-eVIN to 53.5% in the post-eVIN period, reflecting an overall improvement in vaccine management practices. The completeness of data in the vaccine stock registers significantly improved from 29.0% in pre-eVIN to 75.0% in post-eVIN in all eVIN states (p<0.001) (Table 2).

## Vaccine utilization

A significant reduction in the utilization of vaccines was observed after the introduction of eVIN. The maximum saving of doses was seen for Hep-B with a decrease by 77.5%. The reason

**Table 2. Summary of results of indicators evaluated for programmatic assessment of eVIN.**

| Indicators | Sub-indicators | Assessment Indicators | Pre-eVIN | Post-eVIN | % Reduction |
|---|---|---|---|---|---|
| Vaccine Utilization | | Utilization of doses (in Million) | 305.3 | 215.0 | 29.6 |
| Vaccine Stock Management | Stock out | Facilities experienced stock out of vaccines (in %) | 37.8 | 26.3 | 30.4 |
| | Minimum Stock | Facilities observed minimum stock of any vaccines (in %) | 49.8 | 47.8 | 4.0 |
| | Maximum Stock | Facilities observed maximum stock of any vaccines (in %) | 37.4 | 39.2 | -4.8 |
| | Missed Opportunity | Immunization sessions missed due to stock-out and resulting into missed opportunities (in numbers) | | | |
| | | BCG | 2,096 | 1,961 | 6.4 |
| | | HEP-B | 1,837 | 1,833 | 0.2 |
| | | OPV | 13,954 | 8,159 | 41.5 |
| | | DPT | 7,382 | 5,902 | 20.0 |
| | | Penta | 7,093 | 2,109 | 70.3 |
| | | Measles | 5,969 | 3,467 | 41.9 |
| | | TT | 3,729 | 2,593 | 30.5 |
| | Vaccine Stock Updating Duration | Facilities updated vaccine stock daily (in %) | 38.6 | 53.5 | 27.9[a] |
| Vaccine Distribution | Expiry days of vaccines | Mean expiry days | 428 | 384 | 10.3 |
| Vaccine Management | Proper record keeping practices | GoI registers (in %) | 56.2 | 97.4 | 42.3[a] |
| | | Loose papers (in %) | 11.2 | 0.7 | 93.8 |
| Documentation | Vaccine stock register | CCPs with completeness of vaccine stock registers (in %) | 29.0 | 75.0 | 61.3[a] |

[a] is % increase

**Table 3. Vaccine-wise saving of doses.**

| Antigen | Utilization of doses (in Million) | | Saving (%) |
|---|---|---|---|
| | Pre-eVIN | Post-eVIN | |
| BCG | 45.4 | 37.5 | 17.4 |
| DPT | 29.8 | 27.3 | 8.4 |
| Hep-B | 28.9 | 6.5 | 77.5 |
| Measles | 44.0 | 34.7 | 21.1 |
| OPV | 47.6 | 31.1 | 34.7 |
| Pentavalent | 45.5 | 35.1 | 22.9 |
| TT | 64.1 | 42.8 | 33.2 |
| Total | 305.3 | 215.0 | 29.6 |

for the reduction in utilization may be attributed to the limited usage of Hep-B at institutions following the introduction of Pentavalent vaccine. In addition, 34.7% of doses of OPV, 33.2% doses of TT, 22.9% doses of Pentavalent and 21.1% doses of measles vaccines were saved during the post-eVIN period (Table 3).

**State-wise utilization.** The state-wise utilization of vaccines leading to realistic demand reflection of vaccines at the national level. Assam, Chhattisgarh, Himachal Pradesh and Uttar Pradesh have shown more than 30.0% saving of utilized doses. Other states reported marginal saving of doses. However, a marginal decrease of 4.4% and 4.3% has been observed in Madhya Pradesh and Rajasthan respectively (Fig 1).

## Vaccine stock management

**Stock-out of vaccines.** Vaccine-wise analysis for stock-outs showed that all vaccines observed a remarkable reduction in stock-out at facilities across eVIN states (Table 4). Secondary data from UNDP was analyzed for Bihar and Manipur due to under-reporting of data during the visits. Facilities reported stock-out had substantially reduced for all except DPT vaccine (59.8% for OPV, 56.9% for TT, 56.1% for Hep-B and 47.7% for Measles vaccine).

**State-wise stock-out.** Number of facilities having stock-out of any vaccine showed a significant reduction by 30.4% in post-eVIN period (p<0.001). Remarkably not a single instance of stock-out occurred in Chhattisgarh. Out of eight states, five states including Bihar,

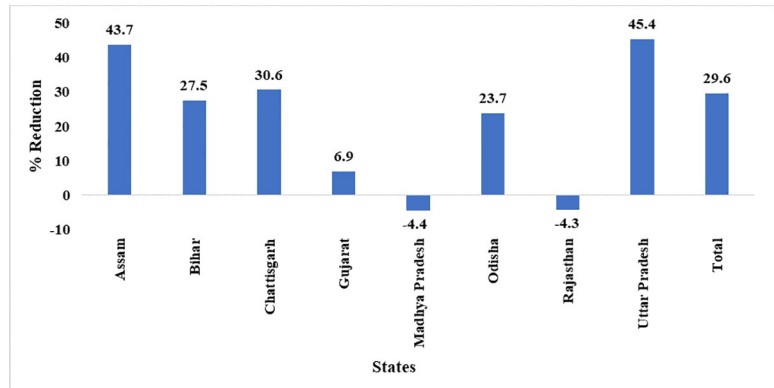

**Fig 1. State-wise reduction in vaccine utilization[a].** [a] Data of Jharkhand, Manipur, Nagaland and Himachal Pradesh are not presented as the number of CCPs from these states are less than 30.

**Table 4. Facilities reporting stock-out of antigens.**

| Antigens | % of facilities | | |
|---|---|---|---|
| | **Pre** | **Post** | **% Reduction** |
| BCG | 10.0 | 7.3 | 27.0 |
| DPT | 14.4 | 13.3 | 7.6 |
| HEP-B | 10.7 | 4.7 | 56.1 |
| Measles | 14.9 | 7.8 | 47.7 |
| OPV | 16.9 | 6.8 | 59.8 |
| PENTA | 8.3 | 4.9 | 41.0 |
| TT | 6.5 | 2.8 | 56.9 |

Chhattisgarh, Gujarat, Rajasthan, and Uttar Pradesh have showed more reduction in stock-out of any vaccine at facility than the overall reduction of 30.4% (Fig 2).

**Minimum stock of vaccines.** The facilities reporting minimum stock of vaccine was more in the case of DPT and Pentavalent compared to any other vaccines. The facilities showing minimum stock of DPT had increased from 32.9% in pre-eVIN to 34.5% in post-eVIN phase (Table 5).

**State-wise minimum stock.** The facilities observing minimum stocks of any vaccine declined in Assam, Bihar, Chhattisgarh, Madhya Pradesh, Rajasthan, and Uttar Pradesh. However, more facilities observed minimum stock in Odisha. There was no change in minimum stock status in Gujarat. Though there was a 4.0% drop in facilities reporting minimum stock of any vaccine after implementation of eVIN (Fig 3).

## Maximum stock of vaccines

The facilities observing maximum stock had increased for TT, Pentavalent, OPV, Measles and BCG, while a decrease is noted for Hep-B and DPT despite the implementation of eVIN (Table 6).

## State-wise maximum stock

Pre- and post-eVIN comparison showed an overall 4.8% increase in maximum stock at facilities. This may be attributed to the increased visibility across all levels of supply chain post eVIN implementation. No change is seen in the facilities observing maximum stock of any vaccine

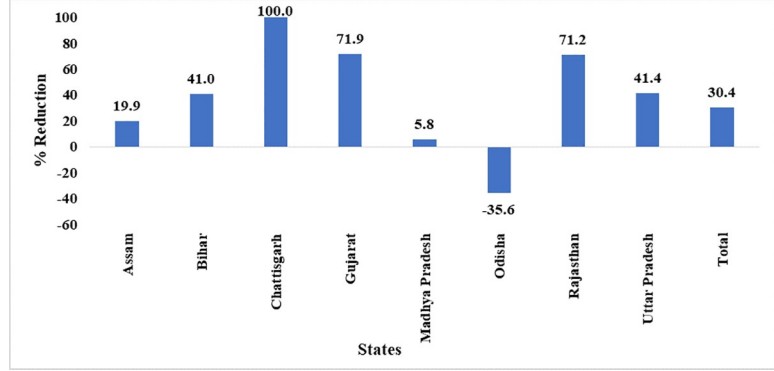

**Fig 2. State-wise reduction in vaccine stock-out[a].** [a] Data of Jharkhand, Manipur, Nagaland and Himachal Pradesh are not presented as the number of CCPs from these states are less than 30.

**Table 5. Facilities reporting Minimum Stock in Pre- eVIN and Post- eVIN period by antigen.**

| Antigens | % of facilities | | |
|---|---|---|---|
| | **Pre** | **Post** | **% Reduction** |
| BCG | 33.4 | 29.8 | 10.8 |
| DPT | 32.9 | 34.5 | -4.9 |
| HEP-B | 19.9 | 15.9 | 20.1 |
| Measles | 31.9 | 27.2 | 14.7 |
| OPV | 36.3 | 31.4 | 13.5 |
| PENTA | 24.6 | 26.4 | -7.3 |
| TT | 29.8 | 17.3 | 41.9 |

especially in Assam, and Chhattisgarh. Most of the facilities in Bihar, Gujarat, Odisha, Rajasthan and Uttar Pradesh showed an increased maximum stock situation in post-eVIN phase.

## Missed opportunity of vaccines

The subsequent number of missed opportunities for all antigens had reduced after post-eVIN implementation, however, the reduction in missed opportunity due to BCG and Hep-B was found to be marginal. The highest reduction of 70.3% in events of missed opportunities was observed for Pentavalent (p<0.001) (Table 2).

## Vaccine distribution management

**Replenishment time.** An attempt was made to assess the impact of eVIN on mean replenishment time and change in replenishment time between supply and indenting of vaccines across cold chain facilities. Mean replenishment time between supply and indent had reduced by 57% across the facilities in 12 states. Overall reduction in replenishment time across facilities was statistically significant (p<0.001). The time between indent and supply of vaccines had decreased for all the states except Jharkhand.

**Order fill rate.** It is observed that overall, there was a marginal increase of 2% in order fill rate from pre- to post-eVIN phase.

**Mean expiry days.** Number of days left in expiry of the vaccine at the level of cold chain points indicated an overall improvement of 10.3% in management of vaccines. Among the 12 states, Uttar Pradesh showed the highest improvement in managing expiry days in post-eVIN

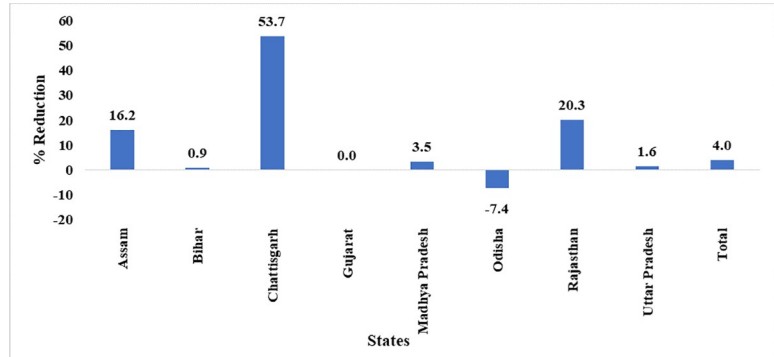

**Fig 3. State-wise reduction in minimum stock-out of vaccines[a].** [a] Data of Jharkhand, Manipur, Nagaland and Himachal Pradesh are not presented as the number of CCPs from these states are less than 30.

**Table 6. Facilities reporting excess stock in Pre- eVIN and Post- eVIN period by antigen.**

| Antigens | % of facilities | | |
|---|---|---|---|
| | **Pre** | **Post** | **% Reduction** |
| BCG | 25.1 | 25.8 | -2.8 |
| DPT | 28.2 | 21.6 | 23.4 |
| HEP-B | 23.7 | 17.2 | 27.4 |
| Measles | 21.4 | 23 | -7.5 |
| OPV | 26.7 | 30.3 | -13.5 |
| PENTA | 21.4 | 27.2 | -27.1 |
| TT | 20.1 | 27.7 | -37.8 |

intervention (Fig 4). Odisha demonstrated a gross reduction in the mean expiry days left for the vaccines. Across all levels of supply chain, adherence to FEFO was observed.

## Vaccine management

**Record keeping practices.** The utilization of the GoI register across the facilities was improved by 42.3% after eVIN implementation. In Chhattisgarh, this utilization was 100.0%. The utilization of Government of India register was encouraging across the facilities in the 12 eVIN states. The usage of loose papers across facilities after the introduction of eVIN was reduced by 93.8%. The practice of using loose papers had stopped in Chhattisgarh, Gujrat, Madhya Pradesh, and Odisha. Usage of loose paper in Bihar was not observed even in pre-eVIN phase. The overall improvement in utilizing the GoI registers and reduction in usage of loose papers across facilities after the introduction of eVIN was statistically significant (p<0.001).

**Vaccine stock updating duration.** During pre-eVIN period, only 38.6% facilities updated vaccine stock on a daily basis, while, during post-eVIN period, 53.5% facilities updated vaccine stock on daily basis. The facilities reported weekly (including daily) updating of stock registers had gone up from 72.5% to 81.1% in the post-eVIN phase.

## Documentation

**Completeness.** The completeness in indent forms had increased from 26.0% in pre-eVIN to 69.0% in post-eVIN (p<0.001). The highest improvement was evident in Nagaland (from

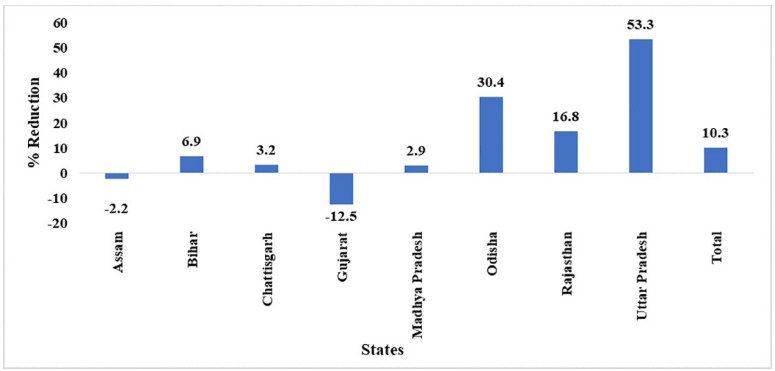

**Fig 4. State wise reduction in mean expiry days left for vaccines[a].** [a] Data of Jharkhand, Manipur, Nagaland and Himachal Pradesh are not presented as the number of CCPs from these states are less than 30.

0.0% to 100.0%) followed by Chhattisgarh (from 0.0% to 96.8%) and Uttar Pradesh (from 48.0% to 95.0%). In the case of vaccine stock registers, data completeness significantly improved from 29.0% in pre-eVIN to 75.0% in post-eVIN in all eVIN states (p<0.001). Completeness of temperature logbook was assessed based on whether temperature was plotted in the morning and evening as mandated, whether defrosting and preventive maintenance records are maintained as mentioned. The completeness of the temperature logbook significantly improved from 28.5% in pre-eVIN to 69.8% in post-eVIN across e-VIN states (p<0.001).

**Accuracy.** Overall, 93.6% match in all eVIN states being observed while comparing the vaccine-wise total stock recorded in stock register with vaccine-wise total stock recorded in eVIN record. Physical stock count of each vaccine kept in ILR and DF at CCPs was done by the field investigator and the findings corroborated with the eVIN record. Overall, 92.0% facilities have shown match of count with eVIN update across all eVIN states.

## Discussion

The vaccine utilization data of Immunization Division, MoHFW reflected savings of approximately 90 million doses of vaccines after eVIN implementation. These savings could be attributed to the roll-out of eVIN across and other initiatives such as the introduction of open vial policy, effective vaccine management assessments, and continuous follow-up with an improvement plan in place. Overall, facilities observing stock-out of any vaccine have significantly reduced after eVIN introduction. But the instances of minimum and maximum stock remained unimproved. Even after reduction in stock-out, 26.3% facilities still observed stock-out in the post-eVIN phase. The variation in mechanism of vaccine distribution may also be affecting stock-outs. The vaccines are being pushed or pulled irregularly on weekly or monthly basis and often depending on the need of the facilities. The analysis of missed opportunities due to reduction in stock-outs revealed that after implementation of eVIN, fewer beneficiaries were getting omitted due to stock-out as compared in the pre-eVIN period. The highest reduction was observed in DPT (reduced by 70.3%), and lowest in BCG (reduced by 6.4%) and almost no change in Hep-B (reduced by 0.2%) after eVIN implementation. Data pertaining to adherence of FEFO (First expiry First out) practices at the level of cold chain facilities and at the level of district vaccine stores was captured in the assessment. The mean number of days (of vaccine expiry) has reduced from 428 in pre-eVIN to 384 in the post-eVIN, at CCP level signifying that the 'First Expiry First Out' (FEFO) is being practiced. The completeness of record keeping was checked against applicable fields of the indent form, register and temperature logbook. Significant improvement in the completeness of records was observed in post-eVIN across eVIN states.

## Conclusion

eVIN has set up a strong example of how technology can be leveraged to enhance efficiency and effectiveness of the public health measures. eVIN system is playing a pivotal role in effective and efficient management of vaccine supply, supervision and monitoring. The findings of this assessment suggest positive changes in the areas of vaccine utilization, stock management and distribution and documentation. States are benefitting with the implementation of eVIN and have been able to improve planning, management of stocks and distribution of vaccines to the last mile.

## Limitations of the study

Limiting the observation period of six months for assessment due to possibility of non-availability of registers in pre-eVIN period beyond this time frame was a challenge. Also, as the

eVIN was rolled out in a phase-wise manner across the states so time period varied among the states for pre-eVIN assessment. Although no foreseen biases were observed but this may cause concern in exact comparison of pre-post period estimates for states.

## Author Contributions

**Conceptualization:** Vandana Gurnani, Prem Singh, Pradeep Haldar, Mahesh Kumar Aggarwal, Pritu Dhalaria.

**Data curation:** Kiran Agrahari.

**Formal analysis:** Vandana Gurnani, Kiran Agrahari, Satabdi Kashyap, Shreeparna Ghosh, Ruma Bhargava, Pritu Dhalaria.

**Investigation:** Prem Singh, Kiran Agrahari.

**Methodology:** Prem Singh, Kiran Agrahari, Pritu Dhalaria.

**Project administration:** Kiran Agrahari.

**Resources:** Pradeep Haldar, Mahesh Kumar Aggarwal.

**Software:** Mrinal Kar Mohapatra.

**Supervision:** Vandana Gurnani, Prem Singh, Pradeep Haldar, Mahesh Kumar Aggarwal, Kiran Agrahari, Pritu Dhalaria.

**Validation:** Satabdi Kashyap.

**Visualization:** Pradeep Haldar, Mahesh Kumar Aggarwal.

**Writing – original draft:** Vandana Gurnani, Satabdi Kashyap, Shreeparna Ghosh, Ruma Bhargava, Pritu Dhalaria.

**Writing – review & editing:** Prem Singh, Satabdi Kashyap, Shreeparna Ghosh, Mrinal Kar Mohapatra, Partha Nandi, Pritu Dhalaria.

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
