## [Decision Letter · Decision Letter 0]

29 Jun 2020

PONE-D-20-02529

Programmatic Assessment of electronic Vaccine Intelligence Network (eVIN)

PLOS ONE

Dear Dr. Dhalaria,

Thank you for submitting your manuscript to PLOS ONE. After careful consideration, we feel that it has merit but does not fully meet PLOS ONE’s publication criteria as it currently stands. Therefore, we invite you to submit a revised version of the manuscript that addresses the points raised during the review process. Please submit your revised manuscript by Aug 13 2020 11:59PM. If you will need more time than this to complete your revisions, please reply to this message or contact the journal office at plosone@plos.org. Please include the following items when submitting your revised manuscript:

We look forward to receiving your revised manuscript.

Kind regards,

Khin Thet Wai, MBBS, MPH, MA (Population & Family Planning Resear

Academic Editor

PLOS ONE

Journal Requirements:

2. Please include additional information regarding the survey or questionnaire used in the study and ensure that you have provided sufficient details that others could replicate the analyses. If you developed and/or translated a questionnaire as part of this study and it is not under a copyright license more restrictive than Creative Commons Attribution (CC-BY), please include a copy, in both the original language and English, as Supporting Information.

3.Thank you for stating the following financial disclosure:

 'The funders had no role in study design, data collection and analysis, decision to publish, or preparation of the manuscript.'

4. We noted in your submission details that a portion of your manuscript may have been presented or published elsewhere. Please clarify whether this conference proceeding or publication was peer-reviewed and formally published. If this work was previously peer-reviewed and published, in the cover letter please provide the reason that this work does not constitute dual publication and should be included in the current manuscript.

Additional Editor Comments:

This is an essential study in support of vaccine supply and logistic management system.

(1) To clarify whether the.larger study has been published or not and if yes please add the citation.

(2) To add limitations of the study in the discussion section.

Reviewers' comments:

Reviewer's Responses to Questions

**Comments to the Author**

1. Is the manuscript technically sound, and do the data support the conclusions?

Reviewer #1: Partly

Reviewer #2: Yes

2. Has the statistical analysis been performed appropriately and rigorously? 

Reviewer #1: Yes

Reviewer #2: Yes

3. Have the authors made all data underlying the findings in their manuscript fully available?

Reviewer #1: Yes

Reviewer #2: Yes

4. Is the manuscript presented in an intelligible fashion and written in standard English?

Reviewer #1: Yes

Reviewer #2: Yes

5. Review Comments to the Author

Reviewer #1: The manuscript is well written on a topic of National relevance. I have a few queries as under:

1. Methods: Please mention the pre-eVIN period. Is it the same or different for different CCPs?

2. There are 9 EVM criteria indicators. Why has only 4 criteria been selected for assessment in this study?

3. How was the design effect of 1.2 taken? How was the intra-class correlation calculated?

4. Please elaborate the sampling strategy

5. How is the data in pre-eVIN phase collected?

6. Data collection method is unclear, as in methods section, mention of Computer Assisted Personal Interviewing (CAPI) has been mentioned whereas in Table 1, under documentation, mention has been made of 'checking completeness of records'

7. The pre-eVIN period of 6 months has been compared with just 3 months of post-eVIN, hence the data of both periods cannot be compared. The reason for this is not understood.

8. Results: Is the data of pre-eVIN and post-eVIN period averaged for a month? As the post-eVIN period is smaller,

and hence 215.0 million doses were utilized as compared to 305.3 million doses utilised in a longer pre-eVIN period. How can the change be attributed to eVIN?

9. Instead of vaccine utilization, vaccine wastage could have been calculated.

10. If there is clustering of births in some months, vaccine utilization may be different, can it be compared?

11. There are other factors which affect vaccine utilization or vaccine stock outs, like difficulty in delivery of vaccine to the CCPs because of difficult road conditions, vaccine drop-outs particularly for Pentavalent vaccine, OPV, etc in the pre-eVIN and post-eVIN period, etc.

12. The labelling of graphs has to be done properly

You are requested to re-look at the above comments and amend your manuscript.

Reviewer #2: Please explain the following points to make the manuscript more informative

The manuscript shall be useful for scaling up the services to the whole of India.

1. Sample size calculation was based on stock out rates. What was the reduction expected (effect size) Design effect 1.2 was considered. Please provide reference for the same.

2. Comparison was made pre eVIN and post eVIN. Six moths reference was taken Pre eVIN and three months post eVIN was compared. Comparison should have been done for similar period/ same months.

3. Major gain was reduction of Utilization Doses. (29.6% reduction of utilization doses). How many beneficiaries received immunization n pre eVIN and Ppost eVIN period.

4. How was states were selected. Expect Himanchal and Gujarat all other states started from weaker health system had poor indicators at the begining.

6. PLOS authors have the option to publish the peer review history of their article (what does this mean?). If published, this will include your full peer review and any attached files.

Reviewer #1: **Yes: **Dr.Paramita Sengupta

Reviewer #2: **Yes: **Binod Kumar Patro

---

## [Author Response · Author response to Decision Letter 0]

11 Sep 2020

Additional Editor Comments:

This is an essential study in support of vaccine supply and logistics management system.

(1) To clarify whether the larger study has been published or not and if yes please add the citation.

Yes, the larger study has been published as a report, but not as a peer reviewed document. The citation of the published report has been added in the manuscript as reference number 12.

(2) To add limitations of the study in the discussion section.

Limitation of the study is included after conclusion section of the manuscript.

Comments to the Author

1. Is the manuscript technically sound, and do the data support the conclusions?

Reviewer #1: Partly

Reviewer #2: Yes

2. Has the statistical analysis been performed appropriately and rigorously?

Reviewer #1: Yes

Reviewer #2: Yes

3. Have the authors made all data underlying the findings in their manuscript fully available?

Reviewer #1: Yes

Reviewer #2: Yes

4. Is the manuscript presented in an intelligible fashion and written in standard English?

Reviewer #1: Yes

Reviewer #2: Yes

5. Review Comments to the Author

Reviewer #1: The manuscript is well written on a topic of National relevance. I have a few queries as under:

1. Methods: Please mention the pre-eVIN period. Is it the same or different for different CCPs?

For Pre eVIN, assessment period was fixed as six months’ time frame before the month of roll-out of eVIN. However, due to roll-out of eVIN in a phase wise manner, the exact six months duration had been different for states and districts. The detail on time of roll-out have been included in the manuscript.

2. There are 9 EVM criteria indicators. Why has only 4 criteria been selected for assessment in this study?

In this study, 5 EVM criterion have been used which are on vaccine utilization, stock management, vaccine distribution, vaccine management practices and documentation. The other indicators on eVIN are human resource and training, vaccine wastage, temperature monitoring and cold chain equipment. The larger study has captured all these domains but due to a different study design or source of information, these indicators have not been mentioned in this manuscript.

3. How was the design effect of 1.2 taken? How was the intra-class correlation calculated?

Design effect of 1.2 has been recommended for facility surveys in developing countries. Reference document is added in the manuscript. The design effects for most of the facility survey estimates of interest is considered to be very low because (1) the list of samples is not clustered at all and (2) both the cluster sizes (that is, number of sample facilities) and the intra cluster correlations in the area sample is small. Therefore, for purposes of calculating sample sizes that the value is about 1.2 at the maximum.

4. Please elaborate the sampling strategy

All eVIN states were selected for pre- and post-design. For the selection of CCPs, two-stage sampling technique was deployed; the first stage for selecting districts, followed by selection of CCPs in the second stage. 

Stage 1: Selection of districts: Within each eVIN state, districts were selected based on Probability Proportional to Size Sampling (PPS). The total number of selected districts in 12 eVIN states was 37. Based on relative proportion of each state, the number of districts to be sampled was: Assam (3), Chhattisgarh (3), Gujarat (3), Jharkhand (2), Manipur (1), Nagaland (1), Odisha (3), Bihar (4), Himachal Pradesh (1), Madhya Pradesh (5), Rajasthan (3), and Uttar Pradesh (8). 

The districts were arranged in ascending order based on the proportionate share of cold chain points out of the total cold chain point in the state. An interval (N/n) factor was calculated by dividing the number of total districts (N) in the region by the number of districts (n) to be selected. After selecting the first district randomly, every (N/n)th district was selected until the required number for districts was obtained. 

Stage 2: Selection of Cold Chain Points: Cold chain points were randomly selected in each of the selected district.

It has been added in the manuscript under “Methods”

5. How is the data in pre-eVIN phase collected?

Data collection for programmatic assessment was carried out between May and July 2018. Quantitative data was collected from the stock registers, vaccine distribution registers, temperature log books and other important registers at CCPs. Completeness and accuracy were assessed through stock registers and eVIN record, eVIN record and physical count. CAPI or “Computer Assisted Personal Interviewing” technique was employed using tablets/mobiles for online data entry. This ensured the quality of data collection and elimination of time involved in data entry. 

This also has been added in the manuscript under “Method”.

6. Data collection method is unclear, as in methods section, mention of Computer Assisted Personal Interviewing (CAPI) has been mentioned whereas in Table 1, under documentation, mention has been made of 'checking completeness of records'

The questionnaire used in the study is now available as supplementary information with the manuscript. All the quantitative information was collected in CAPI. 

To check whether the documentation practices, has improved with the roll-out of eVIN, records were observed to see their completeness and accuracy. The information on completeness and accuracy was quantified into three categories (available in the questionnaire) and the observation was marked in the CAPI in respective question’s reply.

7. The pre-eVIN period of 6 months has been compared with just 3 months of post-eVIN, hence the data of both periods cannot be compared. The reason for this is not understood.

We regret for the error in mentioning the post-eVIN period. The period for post-eVIN was from October 2017 to March 2018 (6 months) for the assessment. Correction has been done in the manuscript.

8. Results: Is the data of pre-eVIN and post-eVIN period averaged for a month? As the post-eVIN period is smaller, and hence 215.0 million doses were utilized as compared to 305.3 million doses utilised in a longer pre-eVIN period. How can the change be attributed to eVIN?

For both the Pre-eVIN and Post-eVIN period, data was collected for 6 months duration separately for each of the phase. There was a typing mistake while mentioning the post eVIN period, we apologies for the same. Since both the periods are of equal length (6 months) therefore the data is comparable. 

There has been huge reduction in vaccine utilization from 305.3 million doses in pre-eVIN period to 215.0 million doses in post-eVIN period across 12 eVIN states, resulting into savings of approximately 90 million doses of vaccines therefore the change can certainly be attributed to eVIN in absence of any other intervention in vaccine supply chain arena. 

9. Instead of vaccine utilization, vaccine wastage could have been calculated.

We agree that the direct presentation of vaccine wastage would have given a clearer picture but unfortunately the vaccine wastage record is highly under-recorded in pre eVIN phase. In the larger study report, the vaccine wastage has been computed using UNDP’s data but not presented in this manuscript due to compatibility issues with the study design. 

Vaccine utilization is a broader variable which includes both usage and wastage so vaccine wastage is indirectly taken care of under vaccine utilization. 

10. If there is clustering of births in some months, vaccine utilization may be different, can it be compared?

The Health Management Information System (HMIS) data of India do not suggest clustering of births in specific months at the state level in these 12 states. The vaccine utilization might get affected at cold chain point level, but it gets nullified at the higher levels like district, state or national level. Therefore the six months observation period each for pre and post eVIN phase can be compared.

11. There are other factors which affect vaccine utilization or vaccine stock outs, like difficulty in delivery of vaccine to the CCPs because of difficult road conditions, vaccine drop-outs particularly for Pentavalent vaccine, OPV, etc in the pre-eVIN and post-eVIN period, etc.

Agree, that the difficult road condition has the potential to affect the vaccine stock-out, but its contribution in total stock-out is very small. Plus the poor road condition is a factor which might be existing in pre as well as and post both time frames.

The vaccine coverage is directly related with vaccine utilization. While planning this study, a matching was done to understand the antigen wise number of beneficiaries in pre and post phase. The analysis suggested that the coverage has gone up in post eVIN phase compared with pre eVIN phase. 

12. The labelling of graphs has to be done properly

There was overlapping in the values and labels which has been corrected and incorporated in the manuscript.

You are requested to re-look at the above comments and amend your manuscript.

Reviewer #2: Please explain the following points to make the manuscript more informative

The manuscript shall be useful for scaling up the services to the whole of India.

1. Sample size calculation was based on stock out rates. What was the reduction expected (effect size) Design effect 1.2 was considered. Please provide reference for the same.

This assessment gives an idea of supply chain and logistics management system with and without eVIN by analyzing the specific indicators in pre and post eVIN phase. This required a given number of cold chain facilities to be representative of pre and post eVIN phase separately. Keeping the study objectives in mind sample size was calculated using a prevalence estimate rather than effect size. 

Reference for design effect is added in the manuscript. A design effect of 1.2 was considered after a thoughtful consultation of published evidence in developing countries.

2. Comparison was made pre eVIN and post eVIN. Six months reference was taken Pre eVIN and three months post eVIN was compared. Comparison should have been done for similar period/ same months.

Apologies for the typing mistake in mentioning the period of post eVIN period. Pre and post eVIN period both were of six months duration for each of the phase. This has been corrected in the manuscript. 

3. Major gain was reduction of Utilization Doses. (29.6% reduction of utilization doses). How many beneficiaries received immunization n pre eVIN and Ppost eVIN period.

The number of beneficiaries who received vaccines in pre eVIN and post eVIN phase is out of the scope of this assessment. The main aim of eVIN roll out was to manage vaccine supply chain and logistics therefore this study focused on this specific aspect. The vaccination coverage is dependent upon several supply and demand side factors including efficient management of eVIN system. The HMIS data provides the number of beneficiaries but it has not been presented due to self-reporting mechanism inherent in it.

4. How was states were selected. Expect Himanchal and Gujarat all other states started from weaker health system had poor indicators at the beginning.

There was no state selection in this assessment. By March 2018 (when this study was planned), eVIN roll- out was completed in 12 states of the country. Therefore all these 12 states were considered for a comparison of indicators in pre vs post eVIN phase.

The state wise rolling out of eVIN was a consensual decision between UNDP and state governments. ITSU had no role in the same.

6. PLOS authors have the option to publish the peer review history of their article (what does this mean?). If published, this will include your full peer review and any attached files.

Do you want your identity to be public for this peer review? For information about this choice, including consent withdrawal, please see our Privacy Policy.

Reviewer #1: Yes: Dr. Paramita Sengupta

Reviewer #2: Yes: Binod Kumar Patro

---

## [Decision Letter · Decision Letter 1]

14 Oct 2020

Programmatic Assessment of electronic Vaccine Intelligence Network (eVIN)

PONE-D-20-02529R1

Dear Dr. Dhalaria,

We’re pleased to inform you that your manuscript has been judged scientifically suitable for publication and will be formally accepted for publication once it meets all outstanding technical requirements.

Kind regards,

Khin Thet Wai, MBBS, MPH, MA (Population & Family Planning Resear

Academic Editor

PLOS ONE

Additional Editor Comments (optional):

Reviewers' comments:

Reviewer's Responses to Questions

**Comments to the Author**

1. If the authors have adequately addressed your comments raised in a previous round of review and you feel that this manuscript is now acceptable for publication, you may indicate that here to bypass the “Comments to the Author” section, enter your conflict of interest statement in the “Confidential to Editor” section, and submit your "Accept" recommendation.

Reviewer #1: All comments have been addressed

2. Is the manuscript technically sound, and do the data support the conclusions?

Reviewer #1: Yes

3. Has the statistical analysis been performed appropriately and rigorously? 

Reviewer #1: Yes

4. Have the authors made all data underlying the findings in their manuscript fully available?

Reviewer #1: Yes

5. Is the manuscript presented in an intelligible fashion and written in standard English?

Reviewer #1: Yes

6. Review Comments to the Author

Reviewer #1: (No Response)

7. PLOS authors have the option to publish the peer review history of their article (what does this mean?). If published, this will include your full peer review and any attached files.

Reviewer #1: **Yes: **Dr. Paramita Sengupta

Prof and Head, Department of Community Medicine and Family Medicine,

AIIMS Kalyani, Nadia, West Bengal

---

## [Editor Report · Acceptance letter]

26 Oct 2020

PONE-D-20-02529R1 

Programmatic Assessment of electronic Vaccine Intelligence Network (eVIN) 

Dear Dr. Dhalaria:

I'm pleased to inform you that your manuscript has been deemed suitable for publication in PLOS ONE. Congratulations! Your manuscript is now with our production department. 

Kind regards, 

on behalf of

Dr. Khin Thet Wai 

Academic Editor

PLOS ONE